# Measuring femoral neck loads in healthy young and older adults during stair ascent and descent

Chen Deng [1]*, Jason C. Gillette [2], Timothy R. Derrick[2]

1 Division of Sport Biomechanics, School of Sport Science, Beijing Sport University, Beijing, P.R China,
2 Department of Kinesiology, Iowa State University, Ames, IA, United States of America

* dengchen@bsu.edu.cn

**Data Availability Statement:** All relevant data are within the manuscript and its Supporting Information files.

## Abstract

Understanding the hip loading environment for daily activities is useful for hip fracture prevention, rehabilitation, and the design of osteogenic exercises. Seventeen older adults (50–70 yrs) and twenty young adults (18–30 yrs) were recruited. A rigid body model combined with a musculoskeletal model was used to estimate lower extremity loading. An elliptical cross-section model of the femoral neck was used to estimate femoral neck stress during stair ascent and descent. Two peaks were identified in the stress curves, corresponding to the peaks in the vertical ground reaction force. During stair ascent, significantly higher tension on the superior femoral neck was found for the young group at peak 1 (young: 13.5±6.1 MPa, older: 4.2±6.5 MPa, p<0.001). Also during stair ascent, significantly higher compression on the posterior femoral neck was found for the older group at peak 2 (young: -11.4±4.9 MPa, old: -18.1±8.6 MPa, p = 0.006). No significant difference was found for stair descent. Components of stress (muscle vs. reaction forces; axial forces vs. bending moments) were also examined for each trial of stair ascent and descent. The stresses and their components provided loading magnitude and locations of higher stress on the femoral neck during stair ascent and descent. Understanding femoral neck stresses may be used to help prevent hip fractures, reduce pain, improve rehabilitation, and design osteogenic exercises.

## Introduction

Fractures in the femoral neck play an important role in morbidity and mortality among older individuals. Worldwide it is estimated that the incidence of hip fractures will rise from 1.66 million in 1990 to 6.26 million by 2050 [1, 2]. Thus, femoral neck fractures will play an integral role in the health of aging population [3–5]. Moreover, the estimated lifetime cost for all hip fractures in the United States and UK are staying high over years [6–9]. These consequences suggest a need to investigate the details of femoral neck stresses so the negative effects of these stresses can potentially be reduced. Understanding the loading environment during daily activities is a prerequisite for recommendations that may decrease the risk of hip fractures, reduce pain, hasten rehabilitation, and allow the design of osteogenic exercises.

The mechanics and the functional anatomy of femoral neck stresses are known in general (**Fig 1**), but the relative contributions to these stress sources are not well understood. Based on

**Funding:** This work was supported by the Chinese Universities Scientific Fund in the form of a grant (No. 2019QD023) awarded to CD.

**Competing interests:** The authors have declared that no competing interests exist.

**Fig 1. The proximal femur with the torso weight ($F_{Torso}$) and the hip abductor muscle force ($F_{HAM}$).** The $F_{HAM}$ was divided into two components: the force component along the long axis of the femoral neck ($F_{HAM\text{-}Axial}$) produced compression for both superior and inferior surfaces; the component perpendicular to the femoral neck long axis ($F_{HAM\text{-}Bend}$) bent the femoral neck in the superior direction.

mechanical analysis, it is assumed that the structure of femoral neck could be such that the hip muscles are required to provide abductor moment to counter the adductor torque caused by weight of the torso during single leg stance. The torso weight vector bends the femoral neck concavely downward, which tends to compress the inferior surface of the femoral neck while placing the superior surface in tension. The hip abductor muscles could counter this tendency, as they produce bending of the femoral neck in a concavely superior direction when activated. In addition to the tendency to bend the femoral neck, the component of the torso weight and the muscles that cross the hip generate axial compression on the entire femoral neck. This axial compression increases the inferior surface compression while decreasing the superior surface tension.

Repetitive or acute injuries may result from fractures due to increased load magnitude or frequency (fatigue fractures) [10], or fractures due to insufficient bone strength (fragility

fractures) [11], or a combination of these factors. Aging individuals often experience a decline of muscle mass, strength, and power [12, 13], which may reduce the ability of the abductor muscles to counter the torso weight and lead to increased femoral neck bending. The resulting change in stress could be a factor that influences the risk of femoral neck injury in the older population.

To examine the loading environment of the femoral neck, instrumented prostheses have been developed that directly record the forces and moments acting at the hip joint. Such a system was used to investigate the joint loading environment of the proximal femur during stair ascent and descent [14–18]. These measurements have only been made on a small number of subjects, and they were atypical since they had undergone hip replacement surgery within 11–31 months prior to the testing. These measurements provide a direct measure of the hip joint forces that are invaluable, but the invasive nature and limited subject pool reduces the practicality of this protocol in most laboratory and clinical settings which studies the healthy population. In this study, multiscale modelling techniques were used to estimate hip joint kinetics during stair ascent and descent (or uphill and downhill) [19–22] with an acceptable accuracy compared with the direct measurements [23], and potentially went further to estimate femoral neck stress or strains [24, 25] for the healthy population in a non-invasive way. In addition, some specific modelling methods allow the sources of the bone stress to be identified (muscle vs. reaction and forces vs. moments) and analyzed, the counteract effect of different stress components could be analyzed and strategies to decrease the total stress on the femoral neck could be explored. In the previous research, the femoral neck stresses and stress source analysis were done for the older population [25] but the differences of stress and its source between older and young population were not conducted yet. The comparison between older and young population for the total femoral neck stresses and the components of the stresses could explain the reason of possible stress change on the femoral neck due to aging and the cause of higher risk of femoral neck injuries.

The purposes of this study were 1) to compare stresses on the femoral neck between young and older adults while traversing stairs, and 2) to analyze the contribution of the components of loading to the stress environment and find if some components could reduce the total stress on the femoral neck. It was hypothesized that stresses on the femoral neck would be higher in older adults than young adults.

## Methods

Modelling procedures were developed and presented in a previous publication [25] with older adults ascending and descending stairs. This study focuses on differences that may exist between young adults and older adults. Sample size was based on data from the previous study and estimated using GPower 3.1. It was determined that a minimum of 34 participants in total would be needed to detect differences in stress between the young and older adult groups with a medium effect size of 0.25 for repeated-measures MANOVA with a power of 0.80. Seventeen older adults and twenty younger adults who were free from lower limb injuries volunteered to participate (Table 1). Before participation, each participant signed a written informed consent document that had been approved by the Iowa State University Human Subjects Review Board.

Anthropometric measurements included height, body mass, segment lengths, segment widths, and segment circumferences of the right lower extremity segment. Eighteen reflective markers were placed on the right lower extremity with a minimum of 3 markers/segment: toe, lateral foot and heel for the right foot segment; anterior/posterior leg for the right leg segment; anterior thigh, right hip for the right thigh segment; left hip, right/left ASIS, right/left PSIS and

**Table 1. Mean (standard deviation) of study participants.**

| | | Number | Age (yrs) | Mass (kg) | Height (m) |
|---|---|---|---|---|---|
| Older | Male | 7 | 60 (6) | 75 (14) | 1.73 (0.05) |
| | Female | 10 | 57 (5) | 67 (8) | 1.67 (0.05) |
| | Total | 17 | 58 (5) | 70 (10) | 1.69 (0.05) |
| Young | Male | 10 | 23 (3) | 80 (14) | 1.76 (0.07) |
| | Female | 10 | 23 (3) | 62 (10) | 1.70 (0.07) |
| | Total | 20 | 23 (3) | 71 (12) | 1.73 (0.07) |

sacrum for the pelvis segment; medial/lateral ankle for both the right foot and leg segments; and medial/lateral knee for both the right leg and thigh segments. All anthropometric measurements and marker placements were performed by the same researcher, who had 3–4 yrs experience as a research assistant on performing these measurements.

A static trial in the anatomical position was collected for each participant to estimate joint center locations, and then medial knee/ankle markers were removed. Five trials of stair ascent and five trials of stair descent (three-step staircase, stair height: 19 cm) were performed for all participants. AMTI force platforms (1600 Hz, AMTI, Watertown, MA) were used to measure ground reaction forces on the two lower stairs. For each trial, participants always initiated stair ascent and descent with the left foot as the first step, then the right foot took the second step and contacted the force platform. Motion data were collected by an 8-camera system (160 Hz, Vicon MX, Centennial, CO).

A low-pass Butterworth filter with a cutoff frequency of 6 Hz was used to filter the ground reaction force and motion data [26]. Only the right limb was analysed. The stance phase cycle for stair ascent/descent was identified from foot contact to toe off on the force platform, and then all the cycles were normalized into a percentage of the stance phase. An inverse dynamics analysis with a rigid body model was used to estimate three-dimensional net joint moments and reaction forces at the ankle, knee, and hip joints. Segment masses, center of mass locations, and moments of inertia were obtained using prediction equations for each participant [27]. Joint moments (in Newton-meters) and reaction forces (in Newtons) were calculated in the global coordinate system and transformed into the coordinate system of the proximal segment at each joint.

The three dimensional segment angles obtained during the trials were used as input to musculoskeletal models of each participant. The models contained 44 lower limb muscles [28] and were used to estimate length and velocity adjusted maximal muscle forces, muscle moment arms and orientations using a custom Matlab script [29]. Static optimization was used to select a set of muscle forces that minimized the sum of the squared muscle stresses [30] and balanced the sagittal plane hip, knee and ankle moments, frontal plane hip moment and the transverse plane hip and ankle moments. Solutions were constrained by the maximal dynamic muscle forces estimated with the musculoskeletal model.

$$\text{minimize} \sum_{i=1}^{44} (F_i A_i)^2 \quad \text{Subject to}: \ r_{ij} \times F_i = M_j, \ 0 \le F_i \le \text{Max dynamic } F_i$$

For the ith muscle: $F_i$ is the estimated muscle force, $A_i$ is the muscle physiological cross-sectional area, $r_{ij}$ is the muscle moment arm for the jth joint moment, and $M_j$ is the jth joint moment.

Three dimensional hip joint reaction forces were summed with muscle forces from muscles that crossed the hip joint to obtain hip joint contact forces (in Newtons). The hip joint contact

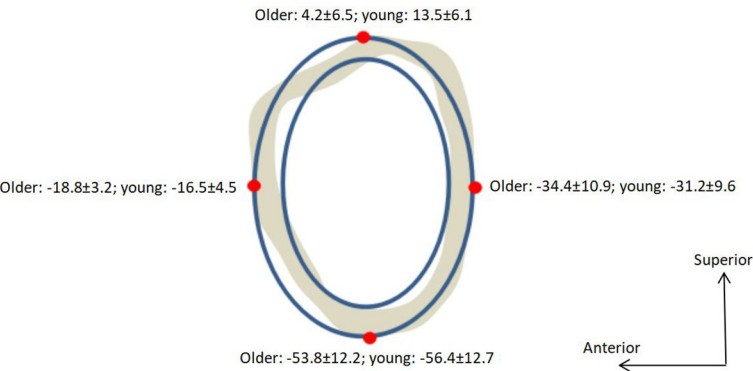

**Fig 2. Elliptical bone model superimposed on a cross-sectional CT scan of the femoral neck, the 4-site stress values (in MPa) were shown during stair ascent at peak 1 for both older and young groups.**

force was then transformed into three forces and three moments acting at the center of the femoral neck. These loads were then applied to a sagittal plane elliptical bone model.

The elliptical bone model (Fig 2) was scaled to the age and gender of the participant. Sub-periosteal width and cortical width data [31] were used to create quadratic prediction equations for the outer and inner diameters along the superior/inferior axis of the femoral neck. The prediction equations adjust for age (20–80 years) and gender:

$$\text{Male Outer Diameter} = -0.0004 \times \text{age}^2 + 0.0962 \times \text{age} + 32.042, R^2 = 0.982$$

$$\text{Male Inner Diameter} = -0.0004 \times \text{age}^2 + 0.1152 \times \text{age} + 27.476, R^2 = 0.987$$

$$\text{Female Outer Diameter} = -0.0004 \times \text{age}^2 + 0.1036 \times \text{age} + 26.662, R^2 = 0.990$$

$$\text{Female Inner Diameter} = -0.0003 \times \text{age}^2 + 0.1102 \times \text{age} + 22.445, R^2 = 0.994$$

where age is in years and diameters are in millimeters. Anterior/posterior diameters were estimated by dividing the superior/inferior diameters with the ratio of maximal to minimal diameters (male: 1.16±0.04; female 1.26±0.03) [32].

Given the forces and moments acting on the centroid of the ellipse, the normal stresses at four locations on the femoral neck periphery were calculated:

$$\sigma_{\text{superior}} = \sigma(-M_{\text{ap}}) + \sigma(F_{\text{axial}}) \quad \sigma_{\text{inferior}} = \sigma(M_{\text{ap}}) + \sigma(F_{\text{axial}})$$

$$\sigma_{\text{anterior}} = \sigma(M_{\text{ml}}) + \sigma(F_{\text{axial}}) \quad \sigma_{\text{posterior}} = \sigma(-M_{\text{ml}}) + \sigma(F_{\text{axial}})$$

where $\sigma_{\text{superior}}$ is the stress on the superior aspect of the femoral neck, $\sigma_{\text{inferior}}$ is the stress on the inferior aspect of the femoral neck, $\sigma_{\text{anterior}}$ is the stress on the anterior aspect of the femoral neck, $\sigma_{\text{posterior}}$ is the stress on the posterior aspect of the femoral neck, $\sigma(M_{\text{ml}})$ is the stress generated by medial-lateral moment, $\sigma(M_{\text{ap}})$ is the stress generated by anterior-posterior moment and $\sigma(F_{\text{axial}})$ is the stress caused by the axial force.

Using this model it is possible to decompose the stress into four sources: muscle forces, muscle moments, joint reaction forces, and joint reaction moments. Reaction forces act at the ellipse centroid, and reaction moments act to bend the femoral neck at the ellipse centroid. This decomposition process has been shown to differentiate between loading associated with stair ascent and descent [25]. The elliptical model has been shown to correlate well with actual

bone models in the tibia. Derrick et al. found that the correlation between peak stress in the ellipse model and peak stress in a bone model generated from CT scans was 0.89 for anterior site tension and 0.96 for posterior site compression [33]. It is believed that the correlation between elliptical model and the actual bone model for the femoral neck should be even better since the shape of the femoral neck is much closer to the elliptical model than the tibia.

Independent variables were age (young vs. older adults) and direction (stair ascent vs descent). The primary dependent variables were stress at two time points (peak 1 and peak 2) and four femoral neck locations (superior, inferior, anterior and posterior). Hip contact forces and hip extensor forces are also presented to help explain and verify the results. The stresses were compared at the two time points during the stance phase that corresponded with the two peak values on the time by stress curves and a cross-correlation was performed to test the similarities of the waveform. The stress on the superior aspect of the cross section did not have a consistent first peak for the older group during stair ascent, so the average time of the peak for young group was used. Positive stress values indicate a tensile stress, and negative values indicate a compressive stress throughout this paper. Comparisons were performed on the absolute value of the stress, making tensile and compressive stress statistically equivalent.

A two-way repeated-measures MANOVA was used to compare the differences between the age groups and the directions, as well as to test for an age by direction interaction (SPSS, IBM Corp). When significant main effects were detected, univariate ANOVAs were performed. The alpha level was set at .05 for these statistical tests. Independent t-tests were used to compare the stresses at the same site between young and older groups. For each half of stance, the peak stresses at each of four surfaces were compared between young and older group, the alpha level was set at .012 for the t-tests to control the familywise errors. If sphericity was violated, a Greenhouse-Geisser correction was performed. Force and moment contributions to the stress were used for explanatory purposes, but not statistically compared.

## Results

The statistical analysis showed that the data satisfied the assumption of normality.

Hip joint contact forces are normalized using the individuals' body weight (BW), then they are presented in the thigh coordinate system and are considered to be acting on the proximal femoral head. The curves tend to be bimodal with peak values at approximately 20 and 80 percent of the stance phase (Fig 3). There was not a significant main effect of age (p = 0.908) or a significant interaction effect of age and direction of travel (p = 0.082).

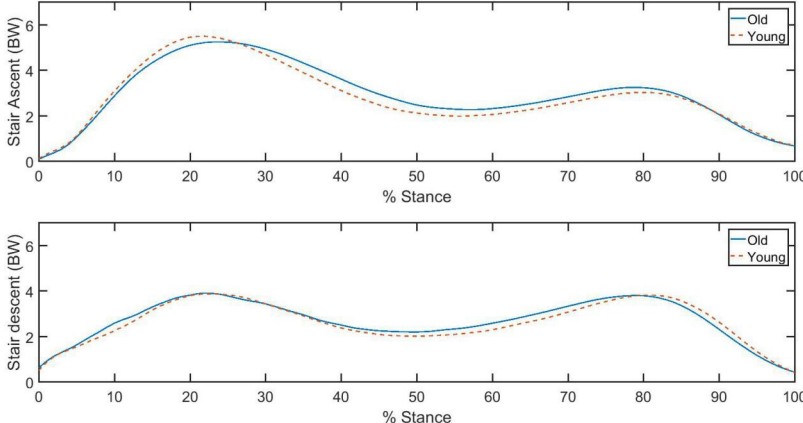

**Fig 3. Ensemble average of hip joint resultant contact forces during stair ascent and descent. BW indicates body weight.**

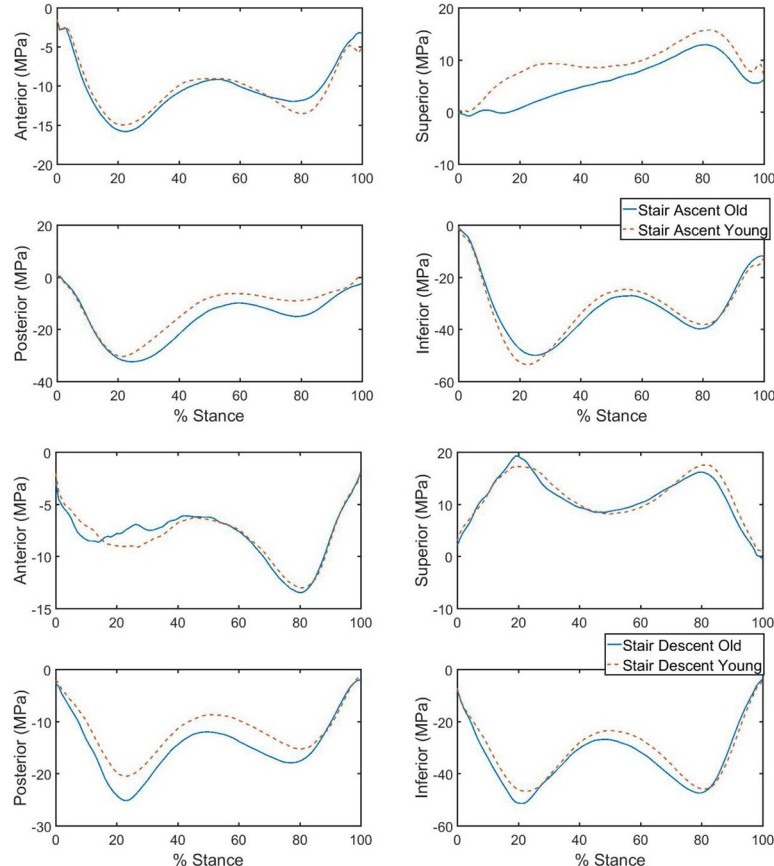

**Fig 4.** Ensemble average of stresses at 4 sites of femoral neck during stair ascent (top) and descent (bottom). Positive values indicate tension, negative values indicate compression.

During both peaks of hip contact force, the dominant loading in the femoral neck was compressive and occurred on the inferior region of the femoral neck during both ascent and descent for both age groups (Fig 4). Peak tensile stress occurred in the superior region. There was a significant main effect of age (p = 0.002), travel direction (p = 0.001), and the interaction between travel direction and age (p = 0.021) on the femoral neck stresses, a-posteriori power analysis was performed and showed the effect size f for MANOVA test was 0.339. Significant differences in stress between young and older groups were found during stair ascent (Table 2) The cross-correlations of the stress curves were high for all the sites: most were ranged from 0.950 to 0.980, and a little bit lower for stair ascent at the superior region (0.901). Peak tensile stress at the superior region of the femoral neck was greater for the young adult group during peak 1 of stair ascent compared to the older adult group (p<0.001, effect size = 1.42). However, peak compressive stress at the posterior region of the femoral neck was greater for the older adult group during peak 2 of stair ascent compared to the young adult group (p = 0.006, effect size = 0.96).

Based on the estimations from the model, the stresses caused by the reaction forces/moments were calculated separately from the stresses caused by the muscle forces/moments so that distinct contributions to the stress load could be assessed, Table 2 showed stress values and Table 3 showed the percentage of stress contributions ($Percentage = \frac{Contribution}{Totalstress} \times 100\%$) from reaction forces/moments and muscle forces/moments. In the equation, the Totalstress

**Table 2. Means (standard deviations) of peak stresses (MPa) in 4 sites of femoral neck for young and older adult groups for stair ascent and descent.**

| Stress Site | Stair Ascent Peak 1 | | | | | | | | | | | | | | |
| --- | --- | --- | --- | --- | --- | --- | --- | --- | --- | --- | --- | --- | --- | --- |
| | Older Adult Stress (MPa) | | | | | | | Young Adult Stress (MPa) | | | | | | | |
| | Muscle | | | Reaction | | | Total | Muscle | | | Reaction | | | Total | |
| | Force Source | Moment Source | Total Muscle | Force Source | Moment Source | Total Reaction | | Force Source | Moment Source | Total Muscle | Force Source | Moment Source | Total Reaction | |
| Superior | -19.8 | -56.7 | -76.5 | -3.4 | 84.1 | 80.7 | **4.2**[1] | -14.3 | -39.2 | -53.5 | -2.6 | 69.6 | 66.9 | **13.5**[1] |
| | (4.6) | (20.0) | (23.8) | (0.6) | (22.6) | (22.1) | **(6.5)** | (5.2) | (12.1) | (15.3) | (0.7) | (19.9) | (19.3) | **(6.1)** |
| Inferior | -21.7 | 54.5 | 32.8 | -3.4 | -83.2 | -86.6 | -53.8 | -19.9 | 47.8 | 27.9 | -3.1 | -81.2 | -84.3 | -56.4 |
| | (4.9) | (20.6) | (17.2) | (0.6) | (22.9) | (23.4) | (12.2) | (3.1) | (15.2) | (13.6) | (0.5) | (20.1) | (20.5) | (12.7) |
| Anterior | -20.0 | 28.0 | 8.0 | -3.0 | -23.0 | -26.0 | -18.0 | -18.8 | 27.3 | 8.5 | -2.9 | -22.2 | -25.1 | -16.5 |
| | (6.2) | (12.2) | (9.0) | (1.1) | (12.1) | (12.7) | (7.9) | (3.2) | (11.4) | (9.9) | (0.5) | (10.4) | (10.5) | (4.5) |
| Posterior | -21.9 | -26.8 | -48.7 | -3.4 | 17.7 | 14.3 | -34.4 | -20.2 | -28.1 | -48.3 | -3.0 | 20.2 | 17.1 | -31.2 |
| | (4.9) | (13.4) | (17.3) | (0.6) | (16.0) | (15.8) | (10.9) | (3.2) | (11.0) | (12.9) | (0.6) | (10.0) | (10.1) | (9.6) |

| Stress Site | Stair Ascent Peak 2 | | | | | | | | | | | | | | |
| --- | --- | --- | --- | --- | --- | --- | --- | --- | --- | --- | --- | --- | --- | --- |
| | Older Adult Stress (MPa) | | | | | | | Young Adult Stress (MPa) | | | | | | | |
| | Muscle | | | Reaction | | | Total | Muscle | | | Reaction | | | Total | |
| | Force Source | Moment Source | Total Muscle | Force Source | Moment Source | Total Reaction | | Force Source | Moment Source | Total Muscle | Force Source | Moment Source | Total Reaction | |
| Superior | -9.9 | -33.9 | -43.8 | -3.0 | 62.6 | 59.5 | 15.7 | -8.6 | -30.4 | -39.0 | -2.8 | 61.6 | 58.9 | 19.9 |
| | (2.1) | (16.4) | (17.7) | (0.8) | (20.0) | (19.3) | (6.1) | (2.2) | (14.5) | (9.8) | (0.5) | (9.5) | (14.0) | (8.4) |
| Inferior | -10.9 | 40.1 | 29.2 | -3.4 | -69.6 | -73.0 | -43.8 | -8.9 | 31.8 | 22.8 | -2.8 | -63.1 | -65.9 | -43.1 |
| | (2.4) | (15.8) | (14.4) | (0.6) | (18.6) | (19.2) | (9.7) | (2.2) | (9.7) | (10.0) | (0.6) | (14.5) | (15.0) | (12.3) |
| Anterior | -9.2 | 5.0 | -4.2 | -2.8 | -7.4 | -10.2 | -14.4 | -8.4 | 4.2 | -4.3 | -2.6 | -9.1 | -11.7 | -15.9 |
| | (2.8) | (9.5) | (8.7) | (0.8) | (11.1) | (11.6) | (7.7) | (2.0) | (10.3) | (10.9) | (0.5) | (10.5) | (10.5) | (6.5) |
| Posterior | -10.6 | -6.3 | -16.9 | -3.0 | 1.8 | -1.2 | **-18.1**[2] | -8.4 | -5.3 | -13.7 | -2.5 | 4.8 | 2.4 | **-11.4**[2] |
| | (2.3) | (9.7) | (10.3) | (0.7) | (11.3) | (11.3) | **(8.6)** | (2.2) | (11.0) | (10.8) | (0.7) | (10.9) | (11.2) | **(4.9)** |

| Stress Site | Stair Descent Peak 1 | | | | | | | | | | | | | | |
| --- | --- | --- | --- | --- | --- | --- | --- | --- | --- | --- | --- | --- | --- | --- |
| | Older Adult Stress (MPa) | | | | | | | Young Adult Stress (MPa) | | | | | | | |
| | Muscle | | | Reaction | | | Total | Muscle | | | Reaction | | | Total | |
| | Force Source | Moment Source | Total Muscle | Force Source | Moment Source | Total Reaction | | Force Source | Moment Source | Total Muscle | Force Source | Moment Source | Total Reaction | |
| Superior | -10.7 | -31.9 | -42.6 | -3.8 | 71.3 | 67.5 | 24.9 | -10.3 | -33.9 | -44.3 | -3.4 | 69.7 | 66.3 | 22.0 |
| | (3.2) | (16.0) | (18.2) | (1.0) | (24.0) | (23.2) | (9.9) | (3.3) | (19.0) | (21.4) | (0.8) | (23.0) | (22.4) | (3.8) |
| Inferior | -13.0 | 36.4 | 23.4 | -4.1 | -76.5 | -80.6 | -57.2 | -11.6 | 38.2 | 26.6 | -3.6 | -74.2 | -77.8 | -51.2 |
| | (3.0) | (18.2) | (16.7) | (0.9) | (25.8) | (26.7) | (15.1) | (3.2) | (19.4) | (17.5) | (0.8) | (23.7) | (24.3) | (10.0) |
| Anterior | -12.2 | -2.4 | -14.6 | -3.1 | 4.8 | 1.7 | -12.9 | -10.4 | -6.5 | -16.9 | -2.7 | 7.2 | 4.5 | -12.5 |
| | (2.7) | (12.5) | (11.5) | (1.1) | (13.9) | (14.6) | (5.4) | (2.7) | (14.7) | (15.7) | (1.1) | (16.4) | (16.1) | (5.5) |
| Posterior | -13.4 | 7.2 | -6.2 | -4.0 | -17.7 | -21.7 | -27.9 | -11.3 | 9.9 | -1.4 | -3.5 | -17.3 | -20.7 | -22.1 |
| | (3.0) | (12.4) | (13.3) | (0.8) | (15.6) | (15.7) | (10.1) | (3.3) | (15.6) | (14.3) | (0.8) | (18.7) | (19.3) | (9.2) |

| Stress Site | Stair Descent Peak 2 | | | | | | | | | | | | | | |
| --- | --- | --- | --- | --- | --- | --- | --- | --- | --- | --- | --- | --- | --- | --- |
| | Older Adult Stress (MPa) | | | | | | | Young Adult Stress (MPa) | | | | | | | |
| | Muscle | | | Reaction | | | Total | Muscle | | | Reaction | | | Total | |
| | Force Source | Moment Source | Total Muscle | Force Source | Moment Source | Total Reaction | | Force Source | Moment Source | Total Muscle | Force Source | Moment Source | Total Reaction | |
| Superior | -13.2 | -15.2 | -28.4 | -2.7 | 49.1 | 46.4 | 18.0 | -11.6 | -14.0 | -25.6 | -2.4 | 47.8 | 45.4 | 19.9 |
| | (3.4) | (10.5) | (12.1) | (0.6) | (18.6) | (18.2) | (8.4) | (2.8) | (10.3) | (11.0) | (0.4) | (15.5) | (15.2) | (7.3) |
| Inferior | -13.9 | 16.2 | 2.3 | -2.8 | -50.5 | -53.3 | -51.0 | -12.1 | 15.3 | 3.2 | -2.5 | -49.4 | -51.8 | -48.6 |
| | (3.4) | (11.3) | (10.3) | (0.6) | (19.3) | (19.8) | (14.4) | (2.7) | (10.4) | (10.3) | (0.4) | (15.3) | (15.6) | (12.5) |

*(Continued)*

**Table 2.** (Continued)

| Anterior | -13.2 | -30.2 | -43.4 | -2.6 | 31.0 | 28.4 | -15.0 | -11.5 | -28.6 | -40.2 | -2.3 | 28.0 | 25.6 | -14.5 |
|---|---|---|---|---|---|---|---|---|---|---|---|---|---|---|
| | (2.8) | (11.8) | (13.3) | (0.6) | (11.6) | (11.4) | (5.3) | (2.6) | (7.9) | (8.9) | (0.5) | (9.6) | (9.5) | (6.2) |
| Posterior | -13.2 | 30.0 | 16.8 | -2.7 | -34.1 | -36.7 | -20.0 | -11.7 | 31.5 | 19.8 | -2.4 | -34.4 | -36.8 | -16.9 |
| | (3.3) | (12.5) | (11.3) | (0.8) | (14.5) | (14.7) | (7.6) | (2.6) | (10.0) | (9.7) | (0.5) | (10.8) | (10.8) | (4.8) |

Values in black were significantly different when comparing the young and older adult groups.

1. $p < 0.001$.

2. $p < 0.01$.

**Table 3. The percentage of stress from each component in 4 sites of femoral neck during stair ascent and descent.**

| Stress Site | Stair Ascent Peak 1 | | | | | | | | | | | | | |
|---|---|---|---|---|---|---|---|---|---|---|---|---|---|---|
| | Older Adult Stress (MPa) | | | | | | | Young Adult Stress (MPa) | | | | | | |
| | Muscle | | | Reaction | | | Total | Muscle | | | Reaction | | | Total |
| | Force Source | Moment Source | Total Muscle | Force Source | Moment Source | Total Reaction | | Force Source | Moment Source | Total Muscle | Force Source | Moment Source | Total Reaction | |
| Superior | -470.3% | -1346.8% | -1817.1% | -80.8% | 1997.6% | 1916.9% | 100.0% | -105.9% | -290.4% | -396.3% | -19.3% | 515.6% | 495.6% | 100.0% |
| Inferior | 40.3% | -101.3% | -61.0% | 6.3% | 154.6% | 161.0% | 100.0% | 35.3% | -84.8% | -49.5% | 5.5% | 144.0% | 149.5% | 100.0% |
| Anterior | 111.1% | -155.6% | -44.4% | 16.7% | 127.8% | 144.4% | 100.0% | 113.9% | -165.5% | -51.5% | 17.6% | 134.5% | 152.1% | 100.0% |
| Posterior | 63.7% | 77.9% | 141.6% | 9.9% | -51.5% | -41.6% | 100.0% | 64.7% | 90.1% | 154.8% | 9.6% | -64.7% | -54.8% | 100.0% |

| Stress Site | Stair Ascent Peak 2 | | | | | | | | | | | | | |
|---|---|---|---|---|---|---|---|---|---|---|---|---|---|---|
| | Older Adult Stress (MPa) | | | | | | | Young Adult Stress (MPa) | | | | | | |
| | Muscle | | | Reaction | | | Total | Muscle | | | Reaction | | | Total |
| | Force Source | Moment Source | Total Muscle | Force Source | Moment Source | Total Reaction | | Force Source | Moment Source | Total Muscle | Force Source | Moment Source | Total Reaction | |
| Superior | -63.1% | -215.9% | -279.0% | -19.1% | 398.7% | 379.0% | 100.0% | -43.2% | -152.8% | -196.0% | -14.1% | 309.5% | 296.0% | 100.0% |
| Inferior | 24.9% | -91.6% | -66.7% | 7.8% | 158.9% | 166.7% | 100.0% | 20.6% | -73.8% | -52.9% | 6.5% | 146.4% | 152.9% | 100.0% |
| Anterior | 63.9% | -34.7% | 29.2% | 19.4% | 51.4% | 70.8% | 100.0% | 52.8% | -26.4% | 27.0% | 16.4% | 57.2% | 73.6% | 100.0% |
| Posterior | 58.6% | 34.8% | 93.4% | 16.6% | -9.9% | 6.6% | 100.0% | 73.7% | 46.5% | 120.2% | 21.9% | -42.1% | -21.1% | 100.0% |

| Stress Site | Stair Descent Peak 1 | | | | | | | | | | | | | |
|---|---|---|---|---|---|---|---|---|---|---|---|---|---|---|
| | Older Adult Stress (MPa) | | | | | | | Young Adult Stress (MPa) | | | | | | |
| | Muscle | | | Reaction | | | Total | Muscle | | | Reaction | | | Total |
| | Force Source | Moment Source | Total Muscle | Force Source | Moment Source | Total Reaction | | Force Source | Moment Source | Total Muscle | Force Source | Moment Source | Total Reaction | |
| Superior | -43.0% | -128.1% | -171.1% | -15.3% | 286.3% | 271.1% | 100.0% | -46.8% | -154.1% | -201.4% | -15.5% | 316.8% | 301.4% | 100.0% |
| Inferior | 22.7% | -63.6% | -40.9% | 7.2% | 133.7% | 140.9% | 100.0% | 22.7% | -74.6% | -52.0% | 7.0% | 144.9% | 152.0% | 100.0% |
| Anterior | 94.6% | 18.6% | 113.2% | 24.0% | -37.2% | -13.2% | 100.0% | 83.2% | 52.0% | 135.2% | 21.6% | -57.6% | -36.0% | 100.0% |
| Posterior | 48.0% | -25.8% | 22.2% | 14.3% | 63.4% | 77.8% | 100.0% | 51.1% | -44.8% | 6.3% | 15.8% | 78.3% | 93.7% | 100.0% |

| Stress Site | Stair Descent Peak 2 | | | | | | | | | | | | | |
|---|---|---|---|---|---|---|---|---|---|---|---|---|---|---|
| | Older Adult Stress (MPa) | | | | | | | Young Adult Stress (MPa) | | | | | | |
| | Muscle | | | Reaction | | | Total | Muscle | | | Reaction | | | Total |
| | Force Source | Moment Source | Total Muscle | Force Source | Moment Source | Total Reaction | | Force Source | Moment Source | Total Muscle | Force Source | Moment Source | Total Reaction | |
| Superior | -73.3% | -84.4% | -157.8% | -15.0% | 272.8% | 257.8% | 100.0% | -58.3% | -70.4% | -128.6% | -12.1% | 240.2% | 228.1% | 100.0% |
| Inferior | 27.3% | -31.8% | -4.5% | 5.5% | 99.0% | 104.5% | 100.0% | 24.9% | -31.5% | -6.6% | 5.1% | 101.6% | 106.6% | 100.0% |
| Anterior | 88.0% | 201.3% | 289.3% | 17.3% | -206.7% | -189.3% | 100.0% | 79.3% | 197.2% | 277.2% | 15.9% | -193.1% | -176.6% | 100.0% |
| Posterior | 66.0% | -150.0% | -84.0% | 13.5% | 170.5% | 183.5% | 100.0% | 69.2% | -186.4% | -117.2% | 14.2% | 203.6% | 217.8% | 100.0% |

The total stress equals the sum of Total Reaction and Total muscle. Total muscle or reaction contribution equals the sum of its force and moment sources.

could be the total stress on each of the four sites on the femoral neck (superior, inferior, anterior and posterior), the input for the contribution could be the muscle or reaction component, or force/moment sources from muscle or reaction (Tables 2 and 3).

The stresses caused by the reaction components were greater in magnitude than the stresses caused by the muscle components on the superior and inferior aspects of the femoral neck cross section. Reaction components caused tensile stresses on the superior aspects and compressive stresses on the inferior aspect of the femoral neck. Stresses were also decomposed according to the contributions from forces and moments. The moment contributions produced much higher stresses than the force contributions (Table 3) for the reaction components (in most femoral neck sites) and the muscle components (in most of superior and inferior sites).

## Discussion

In this study, the femoral neck stress and stress contributions (muscle v.s. reaction; force v.s moment) during stair ascent and descent for older and young adults were estimated, which provides more information about the mechanism of femoral neck stress production for different age groups than the previous work [25]. The hypothesis that femoral neck stresses would be significantly increased in older adults was not supported by most of the stress variables during both stair ascent and descent (Table 2). There was a significant main effect of age and significant interaction between age and travel direction. In contrast to our hypothesis, the peak 1 tensile stress at the superior aspect of the femoral neck was significantly higher in the young adult group during stair ascent. In support of our hypothesis, the peak 2 compressive stress at the posterior aspect of the femoral neck was significantly higher in the older adult group during stair ascent. However, the compressive stresses at the posterior aspect of the femoral neck were much lower than the compressive stresses on the inferior aspect, and therefore are likely less of a concern for injury. The laterally, posteriorly and distally directed hip contact forces were not significantly different between young and old groups during stair ascent and descent, which is similar to previous research [34].

An examination of the stresses caused by the reaction force/moment compared to the muscle force/moment highlights how these relationships affect the total stress environment (Tables 2 and 3; Fig 5). For example, differences in the balance of the stress contributions resulted in the significant difference between young and older adults for peak 1 stress on the superior aspect of the femoral neck. Both the tensile stress caused by reaction moment and the compressive stress generated by the muscle moment were higher for the older adult group. However, these stresses were more closely balanced in the older adult group, resulting in a smaller

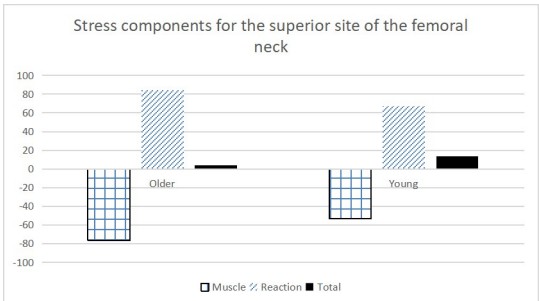

**Fig 5. Components for the peak stress at the superior site of the femoral neck during stair ascent at peak 1.**
Stresses derived from muscle, reaction components and the total stress for each age group were shown.

net tensile stress. Similar patterns were found at both the superior and inferior surfaces of femoral neck: 1) stresses produced by muscle tended to be opposite in direction (tension vs, compression) and smaller in magnitude than the stresses from reaction force, and 2) the greater effect of both muscle and reaction forces was through the production of moments (bending the femoral neck) rather than axial forces. One practical application of these results is that stronger muscles, especially the hip abductors (Gluteus Minimus and Medius etc.), may be effective in reducing the stresses magnitudes on the superior and inferior surfaces of the femoral neck.

For all travel directions and age groups collectively, peak stress values on the femoral neck cross section reached their greatest values on the inferior aspect of the cross section. This was a compressive stress that resulted from summing a high amount of compressive stress due to the reaction moment with a lesser amount of compressive stress due to the muscle force vector. These inferior aspect compressive stresses were not appreciable different between young and older adult groups or stair ascent and descent.

Muscle optimization assumes that the activation of muscles follows the rules of the cost function (minimization of the sum of the muscle stress squared). Differences between the estimated and actual muscle force may occur if a person uses a pattern of recruitment other than the one that is predicted by the cost function. Estimated hip extensor muscle forces were compared to stair ascent and descent EMG data from Hall's research [35] to assess the accuracy of this estimation. S1 Appendix show the comparison between hip extensor muscle forces and EMG activity (including biceps femoris long head, semimembranosus, upper gluteus maximus, and gluteus medius muscles). Even though the EMG activity is a measure of the muscle stimulation and the muscle forces represent the outcomes of muscle activation, the relationship between the EMG and the muscle forces show acceptable agreement during stair ascent. The cross-correlation between muscle force and EMG amplitude were 0.673 for the older adult group and 0.667 for the young group. Similar comparisons between muscle force and EMG activity were shown during stair descent, lower cross-correlations (older: 0.548; young: 0.534) for descent were found. These low cross-correlations may be due to more shifted peaks between the EMG and the muscle forces, which could be explained by 1) the electrical mechanical delay, in which muscle force generation should be shifted back over the excitation signals; and 2) worse co-contraction estimations for the static optimization during eccentric phases of the movement.

There are some limitations in this research, and further improvements could be achieved for the future studies: 1) kinematic data collection using reflective markers on the skin may result in the issues of the soft tissue artefact (STA), techniques to limit these issues should be used in the future analysis; 2) the effect of leg dominance for each participant was not analyzed, further research should consider this factor; 3) the muscle EMG data were not collected from the same participants, the correlation between estimated muscle forces and EMG patterns from the same participants could be more convincing; 4) muscular properties from Arnold's model [28] were based on elderly cadavers, better models for young healthy individuals could ensure the accuracy of estimation; and the analysis for the specific muscle forces could be helpful for the pathological research; 5) individualized femoral neck models with more hip morphology information, especially the models based on the subject specific imaging, could be used to get more accurate estimations. For example, pathological studies that involve bone deformity requires more accurate model for the stress estimation; 6) older population over 70–80 yrs or patients experienced femoral neck fractures may have more issues with BMI loss or muscle strength loss, the findings between young and older (average age as 57 yrs) groups may not be applicable to these populations.

In general, most of the stresses were not different between young and older groups at most sites of the femoral neck: higher tension on the superior femoral neck was found for the young group at peak 1 during stair ascent; during stair ascent, higher compression on the posterior femoral neck was found for the older group at peak 2. Bone stresses and their components (muscle and reaction, forces and moments) were used in the analysis of loading at the femoral neck. The tensile stresses were found at the superior surface, and highest compressive stresses were found at the and inferior surfaces than the anterior or posterior surfaces, and these stresses were predominantly caused by reaction moments and muscle moments. These results could stress the importance of muscle generating moments to balance stresses created by reaction moments for healthy population. This analysis suggested that muscle work may be protective for the femoral neck and stronger muscles may help reduce femoral neck stresses for the healthy older population.

## Supporting information

**S1 Appendix.**
(DOCX)

**S1 Data.**
(XLSX)

## Author Contributions

**Conceptualization:** Chen Deng, Jason C. Gillette, Timothy R. Derrick.

**Data curation:** Chen Deng.

**Formal analysis:** Chen Deng, Timothy R. Derrick.

**Investigation:** Chen Deng, Jason C. Gillette, Timothy R. Derrick.

**Methodology:** Jason C. Gillette, Timothy R. Derrick.

**Software:** Timothy R. Derrick.

**Writing – original draft:** Chen Deng.

**Writing – review & editing:** Chen Deng, Jason C. Gillette, Timothy R. Derrick.

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
