## [Decision Letter · Decision Letter 0]

5 Nov 2020

PONE-D-20-21855

Measuring Femoral Neck Loads in Healthy Young and Older Adults during Stair Ascent and Descent

PLOS ONE

Dear Dr. Deng,

Thank you for submitting your manuscript to PLOS ONE. After careful consideration, we feel that it has merit but does not fully meet PLOS ONE’s publication criteria as it currently stands. Therefore, we invite you to submit a revised version of the manuscript that addresses the points raised during the review process.

We look forward to receiving your revised manuscript.

Kind regards,

Chris Connaboy

Academic Editor

PLOS ONE

Journal Requirements:

Additional Editor Comments (if provided):

Please except our apologies for the delay in getting this first round of reviews completed. We have had a lower than expected acceptance for researchers reviewing manuscripts. Thank you for your patience.

Reviewers' comments:

Reviewer's Responses to Questions

**Comments to the Author**

1. Is the manuscript technically sound, and do the data support the conclusions?

Reviewer #1: Yes

Reviewer #2: Yes

Reviewer #3: Yes

2. Has the statistical analysis been performed appropriately and rigorously? 

Reviewer #1: Yes

Reviewer #2: Yes

Reviewer #3: Yes

3. Have the authors made all data underlying the findings in their manuscript fully available?

Reviewer #1: Yes

Reviewer #2: Yes

Reviewer #3: No

4. Is the manuscript presented in an intelligible fashion and written in standard English?

Reviewer #1: Yes

Reviewer #2: Yes

Reviewer #3: Yes

5. Review Comments to the Author

Reviewer #1: This manuscript examines how femoral stress differs between young and older adults during stair ascent and descent. Further, this manuscript deconstructs these stresses into muscle and force contributions. The authors should be commended for a well-written and interesting manuscript. Each decision on methodological considerations was explained and rationalized well.

General comments:

1. The second paragraph (Lines 42-51) does a good job of explaining the functional anatomy of femoral neck stress. It may be useful to include a figure that depicts the connection between weight, moment, and femoral neck stresses.

2. The paragraph lines 58-72 reads as though a comparison between measurement approaches will be assessed in this study. It is also unclear if the modelling technique to be used in this study provides similar results as the direct measurement approach. This is dealt with later in the manuscript, but is confusing to discuss in the introduction.

3. Lines 68-69: The deconstruction of stress components is briefly mentioned here. It is unclear exactly how each of these components are important and why it is important to examine the components. Please expand on the rationale for Aim 2.

4. Lines 71-71: In line with comment 3, the connection between the components of stress and comparing between older and young individuals is unclear. Please expand on this.

5. Lines 108-109: Were moments and forces normalized to BW? It seems as though they were based on the figures, yet it was not stated. Were these measure normalized before input into further equations?

6. Line 226: Is it possible to add percentage contributions of each stresses/aspects? This could provide values for differences in stress/aspect contribution between young and older individuals. Given contribution differences are discussed multiple times, this could be a great addition to the manuscript.

7. Line 241: thank you for examining why peak 1 superior aspect stress was not consistently present in the older population. Could you provide specific hip muscles that may be the best to target?

Minor comment:

8. Line 154: Typo 'been shown to "be" differentiate'

9. Line 281: Higher stresses compared to what? It is assumed in comparison to the other anterior and posterior surfaces.

Reviewer #2: Thank you for giving me the opportunity to read this interesting manuscript.

The context for the study is clearly outlined and based on appropriate literature.

In the introduction the authors present an argument for comparing older adults with younger adults. The key argument appears to be that decreasing abductor muscle strength in older adults will result in increased femoral bending moments. Surely however it is also possible that decreased muscle forces will lead to decreased hip joint contact forces and therefore decreased overall femoral bending moments. Hip abductor moments must be sufficient to maintain equilibrium or other strategies such as walking aids or the use of handrail on the stairs would be required. The argument therefore for increased femoral bending due to the body weight torque perhaps needs to be further explored.

The methods are clearly described and the statistical methods appear sound.

The results are also clearly outlined and the conclusions drawn from these are reasonable and supported.

There are some potential limitations however which I feel have not been adequately recognised in the discussion.

The 'older' adult sample has an average age of 57 years. Some studies suggest that around 75% of those who experience a hip fracture are over the age of 80 years. Furthermore it has also been suggested that average BMI of those who experience neck of femur fractures is lower than that of this older adult sample. I therefore wonder how generalisable these findings are and whether they can help to further understand the mechanisms of fracture in older adults. Loss of strength appears to accelerate over the age of 60 years and it may be possible that this sample was too young to show the expected differences.

In addition, the issue of hip joint morphology could have been further considered. Some studies have shown an association between hip joint morphology and fracture risk. Whilst it is accepted that is not be ethically acceptable for participants to undergo imaging to gain more accurate anthropometric data, this must be a limitation of the model. Neck of femur fracture may be closely associated with the neck of femur stressors as indicated in this paper however proximal femoral morphology could have a significant impact on the torque around the next of femur. This could have been further explored a a potential limitation of the study.

Reviewer #3: This was an interesting paper with some findings that could potentially have some practical implications. However, this study has certain limitations that similar modelling studies present and also based on the relatively small sample size the findings should be treated with caution.

Regarding the methods, the authors could provide a few more details about the techniques they used to limit the problems with Soft Tissue Artefact and also to mention the experience level of the researcher who undertook those measures. In addition, it would be good to report whether they assessed leg dominance in their sample and why this was not taken into consideration regarding the foot stepping order during the ascent/descent activities. Both of these factors could be potential limitations in this study. Furthermore, although the authors reported the use of a power calculation to estimate the required sample size, it was not clear whether any criterion measures and their characteristics were used so as to preform this calculation. A retrospective power calculation to verify the power of this study to detect stress differences between young/old participants may be worthwhile.

Regarding the statistical analysis, the authors should mention whether the data satisfied the assumption of normality as to use parametric statistics. In addition, they did not report if they used any correction to control for the familywise error due to multiple t-tests.

Finally, main limitations such as the absence of CT scan and EMG data of the participants which could offer more accurate estimates of the musculoskeletal structure and function and also the use of more sophisticated models were discussed. However, in the discussion and the conclusions, any statements about the practical implications and importance of the findings should be more cautious and in light of those limitations.

6. PLOS authors have the option to publish the peer review history of their article (what does this mean?). If published, this will include your full peer review and any attached files.

Reviewer #1: No

Reviewer #2: **Yes: **Judith V. Lane

Reviewer #3: No

---

## [Author Response · Author response to Decision Letter 0]

18 Nov 2020

Hi, there,

As authors, we appreciate all the work the reviewers and editor have done to help us improve the quality of this paper. Here is the rebuttal letter to the reviewers which replies/answers all the questions and concerns.

Again, thank you for providing these comments.

 Best regards,

 Authors for this manuscript

To reviewer #1:

General comments:

1. The second paragraph (Lines 42-51) does a good job of explaining the functional anatomy of femoral neck stress. It may be useful to include a figure that depicts the connection between weight, moment, and femoral neck stresses.

Reply: Thank you for this suggestion, adding a figure can make it easier to understand the explanation.

We added the new figure which depicts the proximal femur and the forces acting on it. The torso weight and the hip abductor muscle force were shown in the figure. The hip abductor muscle force was divided into 2 components: the force component along the long axis of the femoral neck, and component which is perpendicular to the femoral neck long axis.

For details, please take a look at the new Figure 1. Hope this figure will make the explanation much clearer.

2. The paragraph lines 58-72 reads as though a comparison between measurement approaches will be assessed in this study. It is also unclear if the modelling technique to be used in this study provides similar results as the direct measurement approach. This is dealt with later in the manuscript, but is confusing to discuss in the introduction.

Reply: Sorry, we didn’t want to make this paragraph confusing to readers.

The main idea for this paragraph was that the direct measurement is accurate, but due to its low practicality and ethic issues, this method could not be used in the studies with healthy participants. The multiscale modelling techniques could estimate the femoral neck stress in a non-invasive way, which is more suitable to the research for the healthy population.

Duda et al. (2005) showed that the modeling technique could provide the similar results of joint contact force as the direct measurement during walking and stair climbing:

Duda GN, Heller M, Bergmann g. Musculoskeletal loading database: loading conditions of the proximal femur. Theor Issues Ergon Sci. 2005. 6(3-4): 287-292.

Some sentences in this paragraph (line 77-87) were rewrote and the idea that modeling method is more suitable for the healthy population was addressed. The information that modeling technique could provide similar results with the direct measurement and its reference (line 379-380) were added.

3.Lines 68-69: The deconstruction of stress components is briefly mentioned here. It is unclear exactly how each of these components are important and why it is important to examine the components. Please expand on the rationale for Aim 2.

Reply: Sorry, we didn’t explain more about the importance of the stress components.

From the paragraph in line 45-56, analysis of functional anatomy for the femoral neck stress showed that the stress could be generated by muscle forces/moments and reaction forces/moments, these stress components may have the effect to cancel out each other (e.g part of torso weight bending effect could be offset by the hip abductor bending effect). Whether this effect works well will change the total stress for the femoral neck.

In this paragraph (line 83-89), the above thought was addressed.

4.Lines 71-71: In line with comment 3, the connection between the components of stress and comparing between older and young individuals is unclear. Please expand on this.

Reply: Sorry for this unclear description.

For the comparison between older and young population, we examined the total femoral neck stress and the component of the stresses. This analysis could help explore the reason of possible stress change on the femoral neck due to aging and the cause of higher risk of femoral neck injuries.

In this paragraph (line 83-89), the above idea was addressed.

5.Lines 108-109: Were moments and forces normalized to BW? It seems as though they were based on the figures, yet it was not stated. Were these measure normalized before input into further equations?

Reply: Sorry, we didn’t describe the procedures clearly.

Joint reaction moments and forces were not normalized to body weight, but in Figure 3 we reported the normalized resultant hip joint contact forces.

For all the calculations, we kept the original units for the variables (Newtons for forces, Newton-meters for moments) for all the estimation.

Some more explanations were added in line 131, line 146, line 202-203.

6.Line 226: Is it possible to add percentage contributions of each stresses/aspects? This could provide values for differences in stress/aspect contribution between young and older individuals. Given contribution differences are discussed multiple times, this could be a great addition to the manuscript.

Reply: Thank you so much for providing this idea. 

Adding the percentage of stress for each component will be helpful to discuss the difference in the stress contribution. The percentage of contributions was shown in Table 3.

Line 231-234, line 239-240 for the component analysis was rewrote.

7.Line 241: thank you for examining why peak 1 superior aspect stress was not consistently present in the older population. Could you provide specific hip muscles that may be the best to target?

Reply: Thank you, this is a very good point.

We still work on developing our models to figure out , exactly, which muscle forces have “significant effect” on reducing the total stress. But according to the analysis of functional anatomy in the Introduction, hip abductor muscles could be the possible targets for the further analysis. Larger muscles like Gluteus Minimus and Medius may work best, Piriformis, Sartorius and TFL are small and might not be able to produce much force to counteract torso bending effect.

The information that Gluteus Minimus and Medius could be the best to target was added in the discussion, line 273-274.

Minor comment:

8. Line 154: Typo 'been shown to "be" differentiate'

Reply: Sorry for this typo.

We corrected it in line 173.

9. Line 281: Higher stresses compared to what? It is assumed in comparison to the other anterior and posterior surfaces.

Reply: Sorry for the unclear description.

For this paragraph, the tension was found in the superior surface; and inferior surface contains the highest compression comparing with all other surfaces.

This part was rewrote in line 319-321. 

To reviewer #2:

In the introduction the authors present an argument for comparing older adults with younger adults. The key argument appears to be that decreasing abductor muscle strength in older adults will result in increased femoral bending moments. Surely however it is also possible that decreased muscle forces will lead to decreased hip joint contact forces and therefore decreased overall femoral bending moments. Hip abductor moments must be sufficient to maintain equilibrium or other strategies such as walking aids or the use of handrail on the stairs would be required. The argument therefore for increased femoral bending due to the body weight torque perhaps needs to be further explored.

Reply: Thank you for providing this thought.

The argument we addressed in the Introduction is the assumption from the functional anatomy and the mechanics.

We added the figure (Figure 1) to illustrate the mechanics of the femoral neck. For the inferior surface, the greater muscle force and hip contact force would provide greater axial force (along the long axis of the femoral neck) which results in greater compression for the whole femoral neck. The hip abductor moment could bend the femoral neck (results compression for the superior surface of the femoral neck) in the opposite way comparing with the hip contact force (produces tension for the superior surface), the assumption is that the tension in superior surface of the femoral neck could be reduced with greater hip abductor force (both bending moment and axial force work).

The analysis of the stress contribution and percentage of each component (Table 3) could test whether this assumption works for the superior surface of the femoral neck. This idea was added in line 81-89, line 92-93.

To express this idea more like an assumption, we rephrased the paragraph (line 47, 51).

There are some potential limitations however which I feel have not been adequately recognised in the discussion.

1. The 'older' adult sample has an average age of 57 years. Some studies suggest that around 75% of those who experience a hip fracture are over the age of 80 years. Furthermore it has also been suggested that average BMI of those who experience neck of femur fractures is lower than that of this older adult sample. I therefore wonder how generalisable these findings are and whether they can help to further understand the mechanisms of fracture in older adults. Loss of strength appears to accelerate over the age of 60 years and it may be possible that this sample was too young to show the expected differences.

Reply: Thank you for pointing it out.

This is one of major limitations for the method design, so we added this limitation to line 312-314, the paragraph to talk about the limitations.

2. In addition, the issue of hip joint morphology could have been further considered. Some studies have shown an association between hip joint morphology and fracture risk. Whilst it is accepted that is not be ethically acceptable for participants to undergo imaging to gain more accurate anthropometric data, this must be a limitation of the model. Neck of femur fracture may be closely associated with the neck of femur stressors as indicated in this paper however proximal femoral morphology could have a significant impact on the torque around the next of femur. This could have been further explored a a potential limitation of the study.

Reply: Thank you for this comment about the model issues.

Since the simple elliptical model was used in this study, the effect of hip morphology were not considered in the analysis.

In the paragraph talking about the limitations, we mentioned this limitation in point 5 (line 309-312). We rephrased these lines to talk more about the hip morphology issues (line 309-312).

To reviewer #3:

However, this study has certain limitations that similar modelling studies present and also based on the relatively small sample size the findings should be treated with caution.

Reply: Thank you for providing these useful comments and suggestions, these comments will improve the quality of our manuscript.

Detailed replies and changes in the manuscript are listed here: 

1. Regarding the methods, the authors could provide a few more details about the techniques they used to limit the problems with Soft Tissue Artefact and also to mention the experience level of the researcher who undertook those measures.

Reply: Sorry for missing this important information.

The STA issue shouldn’t be ignored due to the skin movement. To reduce the STA effect, we collected the kinematic data (marker data) in static posture at the beginning as part of the calibration process (line 116-117). For each segment, 2-3 or more non-collinear markers were placed on the anatomical landmark and skin (line 108-113). The marker data was processed by the linear interpolation in the analysis, it’s the built in function so we didn’t mention that in the manuscript. 

We know that more procedures are still needed to reduce the error due to Soft Tissue Artefact, so this limitation was discussed in line 302-303.

The researcher performed the measurements was a PhD student (one of the authors), who had experience about 3-4 yrs on doing these measurements. This information was added in line 114-115.

2.In addition, it would be good to report whether they assessed leg dominance in their sample and why this was not taken into consideration regarding the foot stepping order during the ascent/descent activities. Both of these factors could be potential limitations in this study.

Reply: Thank you for this suggestion.

We didn’t analyze the leg dominance for each participant, certainly this is a limitation we should mention in the Discussion (line 303-304).

3.Furthermore, although the authors reported the use of a power calculation to estimate the required sample size, it was not clear whether any criterion measures and their characteristics were used so as to preform this calculation. A retrospective power calculation to verify the power of this study to detect stress differences between young/old participants may be worthwhile.

Reply: Sorry for missing the information of power analysis.

The sample size estimation was performed in Gpower 3.1, the inputs were the power of 0.8 (line 101) and the alpha level as 0.05. The effect size of 0.25 (medium level of effect size) was the default value in the analysis. We supposed that the tested effect size could reach the medium level so we keep this default effect size in the calculation.

In the statistical analysis, if the effect size of 0.25 was not reached with 34 participants, more participants would have been recruited for data collection. 

A retrospective power analysis was mentioned in the beginning of the Result (line 216). This outcome showed that data from 34 participants gives effect size around 0.34, which fulfilled the requirement of the medium effect size in the power analysis. 

4. Regarding the statistical analysis, the authors should mention whether the data satisfied the assumption of normality as to use parametric statistics. In addition, they did not report if they used any correction to control for the familywise error due to multiple t-tests.

Reply: Thank you for this comment and sorry for missing this information.

At the end of Method & start of Result (line 194-197, 201), we mentioned that the data satisfied the assumption of normality and t-test p-values were corrected to control for the familywise error.

5. Finally, main limitations such as the absence of CT scan and EMG data of the participants which could offer more accurate estimates of the musculoskeletal structure and function and also the use of more sophisticated models were discussed. However, in the discussion and the conclusions, any statements about the practical implications and importance of the findings should be more cautious and in light of those limitations.

Reply: You are right, due to the above limitations, the statements about the practical suggestions from the findings should be written with more cautions. 

The statements were rephrased in line 322-325, “could” “may” were used in the statements and the certain population was addressed.

---

## [Decision Letter · Decision Letter 1]

7 Dec 2020

PONE-D-20-21855R1

Measuring Femoral Neck Loads in Healthy Young and Older Adults during Stair Ascent and Descent

PLOS ONE

Dear Dr. Deng,

Thank you for submitting your manuscript to PLOS ONE. After careful consideration, we feel that it has merit but does not fully meet PLOS ONE’s publication criteria as it currently stands. Therefore, we invite you to submit a revised version of the manuscript that addresses the points raised during the review process.

Please address the final comment from reviewer #1

We look forward to receiving your revised manuscript.

Kind regards,

Chris Connaboy

Academic Editor

PLOS ONE

Reviewers' comments:

Reviewer's Responses to Questions

**Comments to the Author**

1. If the authors have adequately addressed your comments raised in a previous round of review and you feel that this manuscript is now acceptable for publication, you may indicate that here to bypass the “Comments to the Author” section, enter your conflict of interest statement in the “Confidential to Editor” section, and submit your "Accept" recommendation.

Reviewer #1: (No Response)

Reviewer #2: All comments have been addressed

2. Is the manuscript technically sound, and do the data support the conclusions?

Reviewer #1: Yes

Reviewer #2: Yes

3. Has the statistical analysis been performed appropriately and rigorously? 

Reviewer #1: Yes

Reviewer #2: Yes

4. Have the authors made all data underlying the findings in their manuscript fully available?

Reviewer #1: Yes

Reviewer #2: Yes

5. Is the manuscript presented in an intelligible fashion and written in standard English?

Reviewer #1: Yes

Reviewer #2: Yes

6. Review Comments to the Author

Reviewer #1: The authors have addressed many of the comments adequately. The reviewers only concern is on the percentage contribution table. It is unclear how >100% contribution could occur. It would be assumed that the total muscle/reaction stress would be 100% or, for example, the muscle force source would add to 100% where each stress site is a contribution of this 100%. Further clarification on how this was calculated is necessary.

Reviewer #2: (No Response)

7. PLOS authors have the option to publish the peer review history of their article (what does this mean?). If published, this will include your full peer review and any attached files.

Reviewer #1: No

Reviewer #2: No

---

## [Author Response · Author response to Decision Letter 1]

4 Jan 2021

To reviewer #1:

General comments:

1.The authors have addressed many of the comments adequately. The reviewers only concern is on the percentage contribution table. It is unclear how >100% contribution could occur. It would be assumed that the total muscle/reaction stress would be 100% or, for example, the muscle force source would add to 100% where each stress site is a contribution of this 100%. Further clarification on how this was calculated is necessary.

Reply: Sorry for the unclear description for the percentage calculation.

In the stress calculation, we calculated the total stress on four sites of the femoral neck separately: superior, inferior, anterior and posterior. For each site, we divided the total stress into several contributions: muscle axial force (along the long axis of the femoral neck), muscle moment (bending effect to the femoral neck), joint reaction force (along the long axis of the femoral neck), and reaction moment (bending effect to the femoral neck).

The percentage calculation was shown as (line 233). The total stress could be the total stress on any one site we defined above, for example, superior site. Then the contribution could be muscle force/moment, reaction force/moment, or the total contribution from muscle or reaction on this site.

The tensile stress was defined as positive and compressive stress was negative. The stress contributions from muscle/reaction or force/moment could be either tensile (positive) or compressive (negative), the total stress was the sum of stress contributions (Table 2). When these stress contributions contain both tensile and compressive stresses (the compensation effect from different contributions could be found in this case), the total stress could be smaller than one or more stress contributions.

If one contribution (e.g muscle axial force contribution) was greater than the total stress, values >100% could be found in the analysis. Moreover, the sign of the contribution and total stress could be opposite (Table 2), so values like -XX% could be found in the Table 3.

Table 3 was revised and more description was added in line 234-237, line 250-252.

---

## [Decision Letter · Decision Letter 2]

6 Jan 2021

Measuring Femoral Neck Loads in Healthy Young and Older Adults during Stair Ascent and Descent

PONE-D-20-21855R2

Dear Dr. Deng,

We’re pleased to inform you that your manuscript has been judged scientifically suitable for publication and will be formally accepted for publication once it meets all outstanding technical requirements.

Kind regards,

Chris Connaboy

Academic Editor

PLOS ONE

Additional Editor Comments (optional):

Thank you for your patience during the review period. All reviews have now been completed.

Reviewers' comments:

Reviewer's Responses to Questions

**Comments to the Author**

1. If the authors have adequately addressed your comments raised in a previous round of review and you feel that this manuscript is now acceptable for publication, you may indicate that here to bypass the “Comments to the Author” section, enter your conflict of interest statement in the “Confidential to Editor” section, and submit your "Accept" recommendation.

Reviewer #1: All comments have been addressed

2. Is the manuscript technically sound, and do the data support the conclusions?

Reviewer #1: Yes

3. Has the statistical analysis been performed appropriately and rigorously? 

Reviewer #1: Yes

4. Have the authors made all data underlying the findings in their manuscript fully available?

Reviewer #1: Yes

5. Is the manuscript presented in an intelligible fashion and written in standard English?

Reviewer #1: Yes

6. Review Comments to the Author

Reviewer #1: (No Response)

7. PLOS authors have the option to publish the peer review history of their article (what does this mean?). If published, this will include your full peer review and any attached files.

Reviewer #1: No

---

## [Editor Report · Acceptance letter]

14 Jan 2021

PONE-D-20-21855R2 

Measuring Femoral Neck Loads in Healthy Young and Older Adults during Stair Ascent and Descent 

Dear Dr. Deng:

I'm pleased to inform you that your manuscript has been deemed suitable for publication in PLOS ONE. Congratulations! Your manuscript is now with our production department. 

Kind regards, 

on behalf of

Dr. Chris Connaboy 

Academic Editor

PLOS ONE